# Excess mortality of infected ectotherms induced by warming depends on pathogen kingdom and evolutionary history

Jingdi Li[1,2]*, Nele Guttmann[3,4], Georgia C. Drew[1,5], Tobias E. Hector[1], Justyna Wolinska[3,4], Kayla C. King[1,2,6]

**1** Department of Biology, University of Oxford, Oxford, United Kingdom, **2** Department of Zoology, University of British Columbia, Vancouver, Canada, **3** Department of Evolutionary and Integrative Ecology, Leibniz Institute of Freshwater Ecology and Inland Fisheries (IGB), Berlin, Germany, **4** Department of Biology, Chemistry, Pharmacy, Institute of Biology, Freie Universität Berlin (FU), Berlin, Germany, **5** Collegium Helveticum, The joint Institute for Advanced Studies (IAS) of the ETH Zurich, The University of Zurich, &The Zurich University of the Arts, Zurich, Switzerland, **6** Department of Microbiology & Immunology, University of British Columbia, Vancouver, Canada

* jingdi.li@biology.ox.ac.uk

**Data Availability Statement:** The data that support the findings of this study are openly available in

## Abstract

Climate change is causing extreme heating events and can lead to more infectious disease outbreaks, putting species persistence at risk. The extent to which warming temperatures and infection may together impair host health is unclear. Using a meta-analysis of >190 effect sizes representing 101 ectothermic animal host–pathogen systems, we demonstrate that warming significantly increased the mortality of hosts infected by bacterial pathogens. Pathogens that have been evolutionarily established within the host species showed higher virulence under warmer temperatures. Conversely, the effect of warming on novel infections—from pathogens without a shared evolutionary history with the host species—were more pronounced with larger differences between compared temperatures. We found that compared to established infections, novel infections were more deadly at lower/baseline temperatures. Moreover, we revealed that the virulence of fungal pathogens increased only when temperatures were shifted upwards towards the pathogen thermal optimum. The magnitude of all these significant effects was not impacted by host life-stage, immune complexity, pathogen inoculation methods, or exposure time. Overall, our findings reveal distinct patterns in changes of pathogen virulence during warming. We highlight the importance of pathogen taxa, thermal optima, and evolutionary history in determining the impact of global change on infection outcomes.

## Introduction

Climate change is resulting in more extreme heating events [1]. Temperature affects all aspects of biology, directly or indirectly impacting the physiology and life-history of all organisms [2]. Typically for ectotherms, environmental warming changes their body temperatures,

Figshare at https://doi.org/10.6084/m9.figshare.22060646.v7.

**Funding:** JW received funding from Beethoven Life-1 (grant no. WO 1587/9-1) grant, funded by the German Science Foundation (DFG, https://www.dfg.de/en). KCK received funding from the Natural Environment Research Council (NE/X000540/1, https://www.ukri.org/councils/nerc/), a Philip Leverhulme Prize, and an NSERC Canada Excellence Research Chair. JL received funding from Pembroke College Oxford Graduate Scholarship. The funders had no role in study design, data collection and analysis, decision to publish, or preparation of the manuscript.

**Competing interests:** The authors have declared that no competing interests exist.

**Abbreviations:** AICc, Akaike's information criterion corrected for small sample size; ES, effect size; GLM, generalized linear model; RR, risk ratio; VIF, variance inflation factor.

influencing their physiological performance and fitness [3]. Shifting global temperature patterns are also changing the geographic distribution of infectious diseases, generating novel transmission opportunities (i.e., host jumping) [4,5].

Positive relationships between warmer temperatures and disease severity (or pathogen virulence) have been observed in a diversity of systems [6–11]. Rising temperature can compromise animal immune responses, leading to increased susceptibility to infections [12–15]. Moreover, by causing stress and disrupting homeostasis, pathogen infection can reduce the upper temperatures tolerated by hosts, contributing to worsened health outcomes under warming [16]. Temperature can also directly mediate pathogen virulence [17–21]. Higher temperatures can up-regulate the expression of virulence factors in the coral pathogen *Vibrio coralliilyticus* [20]. For the human pathogen *Shigella sonnei*, higher temperatures enhance its immune evasion abilities by increasing virulence protein synthesis [21]. Warming may also enhance pathogen metabolism, resulting in faster within-host growth and greater host harm. Such virulence-enhancing effects might be especially true for many bacterial pathogens, as their population growth rates can increase exponentially with temperature, up to their thermal limits [22]. Whether warmer temperatures result in sicker hosts across host–pathogen systems remains unclear [23,24]. In some host–pathogen systems, virulence is unaffected or even reduced by warming [23–28], emphasizing the complex interactions between temperature and disease outcomes.

Both host and pathogen responses to temperature change depend on their thermal tolerance. This trait is characterized by the critical thermal maximum, minimum, and optimum—the upper, lower, and optimal temperatures at which the organism can function [3,29,30]. A mismatch in host and pathogen thermal optima can shape infection outcomes. Cold-adapted pathogens might be less harmful to warm-adapted hosts at higher temperatures [31,32]. Also, eukaryotes generally have a lower thermal tolerance compared to prokaryotes [33], a difference which might make eukaryotic pathogens (e.g., fungi, nematodes) more sensitive to warmer temperatures.

Understanding the broad ecological consequences of elevated temperature on pathogen virulence (or infection-induced host death herein) is crucial for projections of future wildlife disease dynamics and species persistence. If higher temperatures generally exacerbate the severity of infection, climate change could worsen disease outcomes for wild animal populations. Forecasts of species persistence in a changing world might need to account for rates of infection and disease severity [8,34]. However, if virulence is broadly unaffected or context-dependent [23], the health risks posed by warming and infectious disease might need to be evaluated on a case-by-case basis. Any general relationship between warming and virulence would have profound consequences for animal conservation [34,35] and potentially human health [4,36].

In this study, we conducted a meta-analysis to investigate the generality of the relationship between warming and pathogen virulence. We searched the published literature for experimental studies and included a breadth of ectothermic animal host–pathogen systems. We used the most widely reported quantitative form of virulence: pathogen-induced host death [37]. We focused on host death under infection and warming as this measure is pertinent to projections of species extinction and biodiversity loss [38]. We calculated effect sizes via relative risk ratios of death at experimentally high and low temperatures. We separately collected thermal optima data for all included pathogens. We tested whether biological traits (e.g., host type, pathogen type, host–pathogen evolutionary history, pathogen thermal optimum), the scale of temperature increase, and experimental variables (pathogen inoculation method, pathogen dosage, infection exposure/measurement time) could explain variation in the temperature-virulence relationship.

## Materials and methods

### Literature search and data collection

A first-round literature search was performed in Leibniz Institute of Freshwater Ecology and Inland Fisheries (IGB, Germany) in March 2021 on Web of Science Core Collection database. We used search query ['parasit*' OR 'pathogen*' AND 'temperature' AND 'virulen*'] within "Topic" field (encompassing title, abstract, keyword plus, and author keywords), to identify studies reporting the effect of temperature on pathogen virulence. Then a second-round search using the same keywords was conducted in University of Oxford in April 2022 to update the data set. We used host mortality to quantify pathogen virulence. We excluded studies on parasitoids as host mortality under infection would always be 100%. Our main inclusion criteria were: (i) the pathogen was tested in living animal hosts rather than on cells or host tissues; (ii) more than 1 temperature was used in host infection experiment and temperature was the only changing variable in the tested comparisons; (iii) pathogen virulence was measured by assessing host mortality rate under different temperatures; (iv) host mortality data (mean mortality rate and sample size, where sample size is defined as the number of individual host used in each infection treatment) were available publicly or shared by the authors by May 2022.

Following our inclusion criteria, we retrieved data from a total of 60 research articles (see PRISMA flow-chart in S1 Fig and PRISMA-EcoEvo Checklist in S1 Text), published between 1996 and 2021 (summary of studies and moderator variables are provided in S1 Table). In individual articles, data on host mortality were extracted manually from main text, tables, supplementary information, from figures by using WebPlotDigitizer [39], or raw data made available from the authors by request. To maximize the ecological relevance, we imposed a series of restrictions to exclude data from extreme high or low temperatures and extreme high pathogen dosages that are unlikely to happen in nature. Firstly, if more than 5 temperatures were tested, data were extracted from the second lowest and second highest temperatures (when between 2 to 4 temperatures were tested, data were extracted from the highest and lowest temperatures). Secondly, if multiple pathogen dosages were used, data of the middle value (or middle minus one value, in case of even number) was extracted. Thirdly, host mortality data was extracted from the end time point of the assay. However, if mortality began to appear in controls, data were extracted from the time point with control mortality close to or at zero. We excluded certain experimental conditions to capture the most substantial changes in host fitness by selecting conditions with the largest temperature changes, while avoiding the confounding effects of extreme temperatures and dosages. In studies that included multiple temperature levels, the extreme lowest and highest temperatures often represent conditions that are rarely encountered by hosts and pathogens in natural settings, typically resulting in host death or loss of pathogen infectivity confounding the infection effects. We ensured that the exclusion of all these extreme treatments were based on the following criteria: (i) host mortality was primarily caused by temperature rather than infection; (ii) no pathogen growth or infectivity at these extreme temperatures; and (iii) the original study explicitly showed that the extreme temperatures were outside the realistic range for the host–pathogen system (details in S2 Table). As a result, we focused on infection-induced host mortality, rather than death due to temperature alone. Hosts not exposed to pathogens had no mortality (e.g., Jiang and colleagues [7]), very low mortality (<10%, e.g., Brand and colleagues [10]), or host mortality in the infection treatment was corrected for control mortality following Abbott's (1952) formula (e.g., Bugeme and colleagues [11]). Fifteen effect sizes from 6 studies were discarded due to large host mortality (>10%) in control groups.

When host mortality was expressed as a percentage, we estimated the number of alive and dead individuals in both temperature groups by multiplying percentage of host mortality by

the number of host individuals tested. We otherwise obtained the number of alive and dead hosts from the raw data sent by authors. These data were used to calculate the risk ratio (RR) effect size, using *escalc* function in "metafor" R package [40]. RRs represent the relative risk of warming on infected hosts' survival. An overall total of 192 effect sizes from 60 studies were used across 56 pathogens and 50 host species (101 host–pathogen combinations, S2 Fig).

## Moderator variables

To evaluate the impact of various factors on pathogen virulence in response to rising temperatures, we included several moderators in our analysis, including host, pathogen, and experimental variables. To explore whether pathogen type could predict effect size estimates, we recorded whether the pathogen used in the study was bacteria, fungi, nematode, virus, or protist. As the number of effect sizes in protist was low (1 ES for protist), our analysis focused on bacteria, fungi, nematode, and virus pathogens.

We collected data on the optimal growth temperature ($T_{opt}$) for well-studied pathogens as virulence can correlate with pathogen growth rates [41]. These data were collected for 39 pathogen species included in the meta-analysis from published literature (S3 Table, $T_{opt}$ for in vitro growth/fitness was measured for nonviral pathogens). We ordered the experimental temperatures from low to high. Using known thermal optima for these pathogens, we defined whether experimental warming was towards or away from their $T_{opt}$. "Towards $T_{opt}$" was used when the temperature range in the study was below $T_{opt}$. "Away from $T_{opt}$" was used when the study temperature range was above $T_{opt}$. When the low and high temperatures in the study straddled $T_{opt}$, we used "Range includes $T_{opt}$". Specific details on pathogen acclimation to their thermal optima prior to the experiments were generally not provided within individual articles, although pathogens were shown to be cultured within their known thermal ranges.

We tested whether infection outcomes at increased temperature depended on host–pathogen evolutionary history. We defined a moderator based on whether the interaction was "established" (system originally collected from the wild and with shared evolutionary history, see Bally and Garrabou [42] as an example), "semi-established" (host infected by the pathogen in the wild, but combination in the study were not co-isolated, e.g., Wekesa and colleagues [43]), and "novel" (no known record of an infection between the pathogen and the host in nature, e.g., Ekesi and colleagues [44]). The "novel" associations might represent opportunistic infections or a host shift for the pathogen.

We also tested whether host traits—including host life-stage ("adult," "larva," "pupae," and "juvenile") and host type with different immune complexity ("vertebrate" or "invertebrate")—were relevant moderator variables. The availability of host thermal optima data was limited, as well as information on whether hosts were acclimated to their thermal optima or other environments prior to the infection experiment. In most instances, hosts were either recently collected from the field or reared in the laboratory for multiple generations. To account for the potential acclimation effects, we included "host source" as a moderator, distinguishing between wild-collected and lab-reared hosts. Lastly, we collected and evaluated experimental variables, including temperature span (difference between high temperature and low temperature used in the study), infection exposure duration (measured in days), pathogen dosage (within each pathogen type), and pathogen inoculation method ("injection" or "not injection"). These experimental protocols were evaluated given their potential for a bias in extrapolations to nature [45].

## Statistical analysis

To assess the influence of sample size and correction methods on effect sizes, we fitted sample size or correction method as a moderator, and checked that neither sample size nor the

methods used to correct for control mortality influenced the effect sizes (sample size $p = 0.9630$, correction method $p > 0.52$ for all levels). All categorical moderators were assessed for multicollinearity by calculating the variance inflation factor (VIF, *VIF* function in "regclass" R package) when fitted together in a regression model and no multicollinearity was observed (S4 Table, GVIF^(1/(2*Df)) < = 2.5 for all moderators included). Correlation of continuous variables were assessed for Pearson correlation (cor.test function in R) and the possibility of multicollinearity was excluded (S4 Table, $p > 0.05$ for all pairwise correlations).

We coded the data accordingly with RR>1 (logRR>0), indicating greater host mortality at higher temperatures (higher risk associated with higher temperatures), or RR<1 (logRR<0) with lower host mortality at higher temperatures. No relationship between temperature and virulence was indicated by RR = 1 (logRR = 0). Summary effect sizes were calculated by establishing a multilevel random effects meta-regression model using *rma.mv* function in "metafor" R package. To account for taxonomic non-independence, we included species-level random effects for both hosts and pathogens. These random effects were correlated according to a taxonomic correlation matrix derived from a taxonomic tree obtained from the NCBI Taxonomy common tree [46]. When species-level taxonomy was unavailable, genus-level information was used instead. A taxonomic tree was used instead of a phylogenetic tree because, in many cases, species-level phylogenies could not be reliably resolved from the published literature, and substituting species with their relatives would likely introduce bias. Additionally, we included non-taxonomic species-level random effect to capture heterogeneity in effect sizes arising from differences between species that are not associated with evolutionary relatedness. This approach was initially proposed by Schmidt and colleagues [47] and has shown to be a best practice by Cinar and colleagues [48]. To address statistical non-independence due to repeated measures within studies, we included study ID as a random effect in the model.

To determine the importance of moderators, we first fitted a full single joint random effects model without interaction terms, with effect sizes as the response variable. Model selection was performed using *glmulti* function in glmulti R package, with the *fitfunction* customized to implement the multilevel meta-analytic random-effects model. We selected the best-fitting models using Akaike's information criterion corrected (AICc) for small sample size. Models with a ΔAICc < 2 were considered to have substantial support and were included in the final model set. We performed model averaging across this set of top models to account for model uncertainty by calculating the relative importance of each moderator. Additionally, model-averaged estimates and their 95% confidence intervals were obtained for each moderator to assess their significance. Moderators with low importance values (model-averaged importance <0.25 and 95% CI included zero) were considered to have little support for inclusion in the model.

The remaining moderators were fitted in a joint model and interaction terms were included. Similarly, model selection was performed, and the top model set (ΔAICc < 2) was obtained. Since model averaging is generally not recommended with interactions in the model [49], the best model with the simplest structure and the lowest AICc was reported in the main text. The alternative top models can be found in S4 Table. The significance of a moderator was determined based on its consistent significance across all top models. To maximize the number of retained effect sizes, moderators including pathogen thermal optima, infection dosage, and exposure time, which had missing data in some studies, were not included in the single joint model. Instead, multiple regression models were employed to explore the heterogeneity they explained. For multiple regression models, both taxonomic and non-taxonomic structures for host and pathogen species were included as random effects, as well as Study ID. Interactions between these moderators were tested based on a priori hypotheses (all hypotheses and modeling results in S4 Table). Lastly, host mortality data was fitted in generalized linear model

(GLM) with binomial distribution to compare host baseline mortality among host–pathogen systems with different evolutionary histories.

## Results and discussion

### Increased host mortality due to bacterial infections at elevated temperatures

We firstly tested whether higher temperatures generally increased host mortality during infection (i.e., the proportion of dead individuals in the infected host population). A total of 1,649 studies were identified and screened through Web of Science. After title and abstract screening, 1,363 studies were excluded. We then assessed 286 articles in full text, of which 219 were excluded based on our criteria. Ultimately, 67 studies met all the inclusion criteria, having conducted infection experiments at different temperatures. However, 7 of these studies were excluded due to data unavailability despite attempts to contact the authors. As a result, data from 60 studies were included in the final analysis (S1 Fig). These studies included 50 ectothermic animal host species and 56 pathogen species, resulting in 101 host–pathogen combinations (see S2 Fig), and provided 192 effect sizes (ESs). Our data set contained a variety of pathogen types including bacteria, fungi, nematodes, and viruses from diverse geographic regions (S3 Fig). We included diverse host–pathogen systems spanning terrestrial (42 ESs) and aquatic habitats (150 ESs). Terrestrial hosts were predominantly Insecta (125 ESs), and aquatic hosts were predominantly fish (17 ESs) and mollusca (14 ESs) (S2 Fig). Notably, bacterial pathogens were predominantly studied in aquatic animals, while fungal and nematode pathogens were mainly tested in insect hosts (S2 Fig).

Under warming, uninfected host mortality across studies remained zero or low ($<10\%$). Infected hosts did not suffer significantly greater death rates with increase in temperature (Fig 1A, summary RR = 1.64, 95% CI [0.91, 2.95], $p$ = 0.1004). Given the substantial heterogeneity among the effect sizes ($I^2$ = 98.74, $p < 0.0001$), we investigated whether biological and abiotic factors contributed to the observed variation in effect sizes. We specifically tested whether the relationship between temperature and virulence varied by (i) pathogen types (bacteria, fungi, nematoda, viruses); (ii) host–pathogen evolutionary history (established versus novel); (iii) host types and immune complexity (vertebrate versus invertebrate); (iv) host life-stage (e.g., adult versus larva); (v) wild-collected versus lab-reared hosts; (vi) pathogen inoculation method (injection versus not injection); and (vii) the span of temperature changes.

The best model incorporated fixed effects of pathogen type, host–pathogen evolutionary history, host life stage and temperature changes, along with interactions between host–pathogen evolutionary history and pathogen type, as well as between host–pathogen evolutionary history and temperature changes. We found that the impact of higher temperatures on host mortality varied depending upon the type of pathogen infecting. Effect sizes calculated from bacterial pathogens were larger than from viruses (virus es = −3.0, 95% CI [−4.58, −1.41], $p$ = 0.0002). Subgroup analyses showed that among all types of pathogens, only bacteria exhibited a significant positive summary effect size (Fig 1B, bacteria RR = 2.25, $p$ = 0.0002; fungi RR = 0.96, $p$ = 0.8153, nematoda RR = 1.61, $p$ = 0.0716, virus RR = 1.28, $p$ = 0.6868). As most of the effect sizes for host–bacterial systems were calculated from aquatic hosts, our results suggest that bacterial pathogens might pose significantly greater risk during warming, particularly to aquatic animal species. Increased disease outbreaks for aquatic animals, especially fishes, has been an overwhelming issue for the aquaculture industry, causing considerable economic damage and threats to human health [50]. Bacterial diseases are common in aquaculture. We suggest that future disease management should also consider the aggravating effects of warming on the severity of these epidemics.

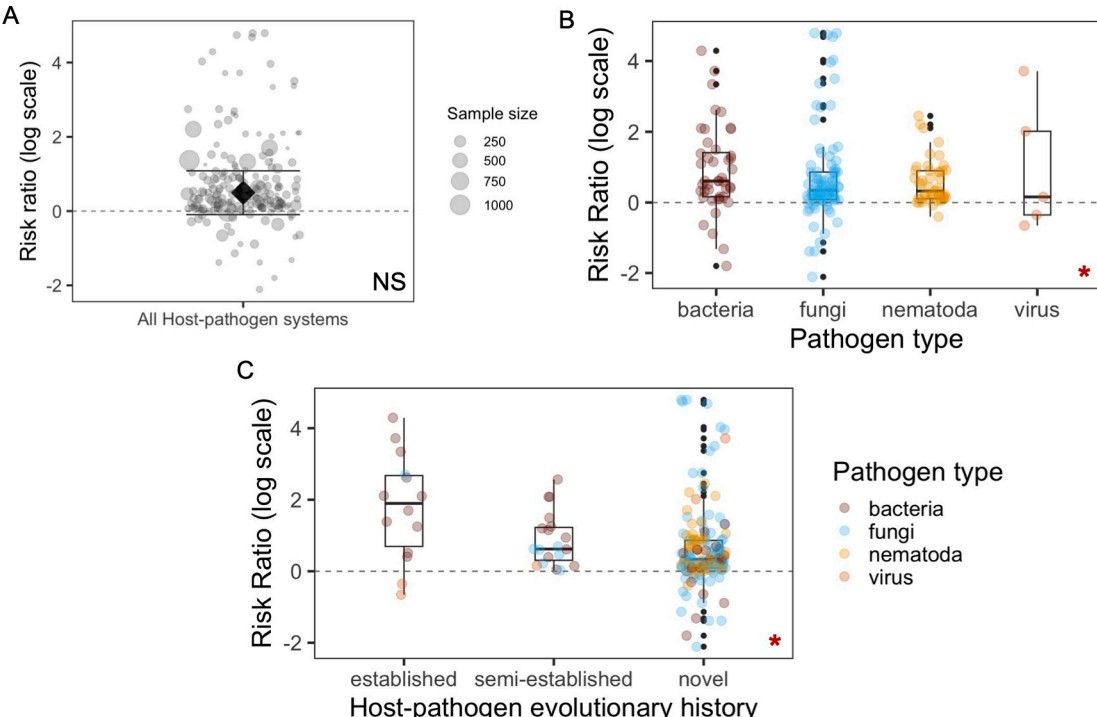

**Fig 1. Results of the meta-analysis on the impact of warming on pathogen-induced host death.** (A) Summary ES (RR) overall is not significantly >1 (logRR>0, indicating positive relationship between temperature and virulence). Bold black diamond with 95% confidence interval represents summary ES, and individual effect size are displayed as jittered points. Point size indicates sample size (number of host individuals in the infection treatment) used for effect size calculation. Effect sizes differed by (B) pathogen type and (C) host–pathogen evolutionary history. Star shown on the bottom right corner indicate significance of moderators. In (B) and (C), jittered points are colored by pathogen type. The data and code needed to generate this figure can be found in https://doi.org/10.6084/m9.figshare.22060646.v7. ES, effect size; RR, risk ratio.

Our results showed that warming increased host mortality to a larger extent in established host-pathogen systems, compared with semi-established or novel systems (Fig 1C, semi-established es = −4.31, 95% CI [−6.84, −1.77], *p* = 0.0009; novel es = −4.03, 95% CI [−6.49, −1.58], *p* = 0.0013). Interestingly, in novel systems, the average host mortality was already higher even at low/baseline temperatures (Fig 2A, novel versus established es = 1.56, *p* = 0.0378), but not at high temperatures (Fig 2B). This high level of virulence might be due to pathogen maladaptation during host shifts [51], suggesting that emerging infections could be highly harmful at present temperatures, and less likely to be worsened during global climate change. On the contrary, the increased severity of endemic infections should be considered when predicting wildlife health under warming. We further showed that the lack of overall virulence-enhancing effect by warming for fungal pathogens was not because they caused higher host mortality at baseline temperatures, though fungal pathogens were mostly from novel systems (S4 Table, *p* = 0.1308). We also observed a significant interaction between pathogen type and evolutionary history (novel:virus es = 5.08, 95% CI [2.38, 7,78], *p* = 0.0002). Given that novel viral infections were represented by only a few effect sizes (*N* = 2) in our data set, generalizing this interaction effect requires caution.

We found that the effect of warming on host mortality was mediated by the extent of temperature change in novel host–pathogen systems. The larger the temperature change, the higher the difference in host mortality between experimental temperatures. A significant interaction was observed between host–pathogen evolutionary history and the span of temperature

## A. Low/baseline temperatures

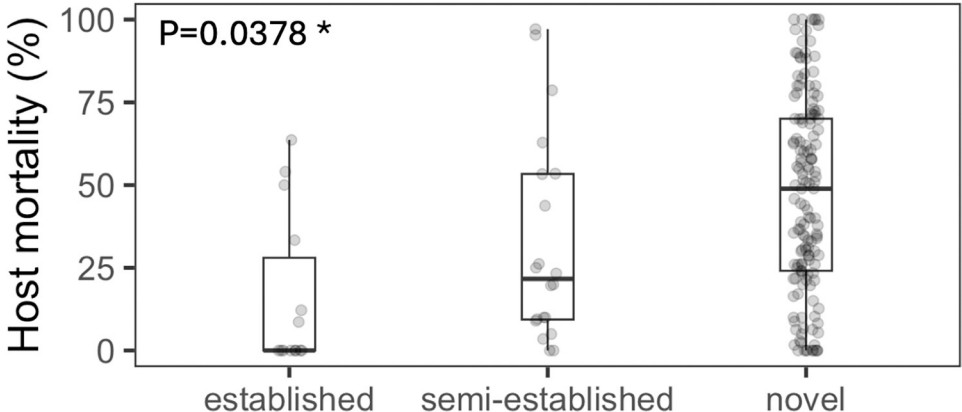

## B. High temperatures

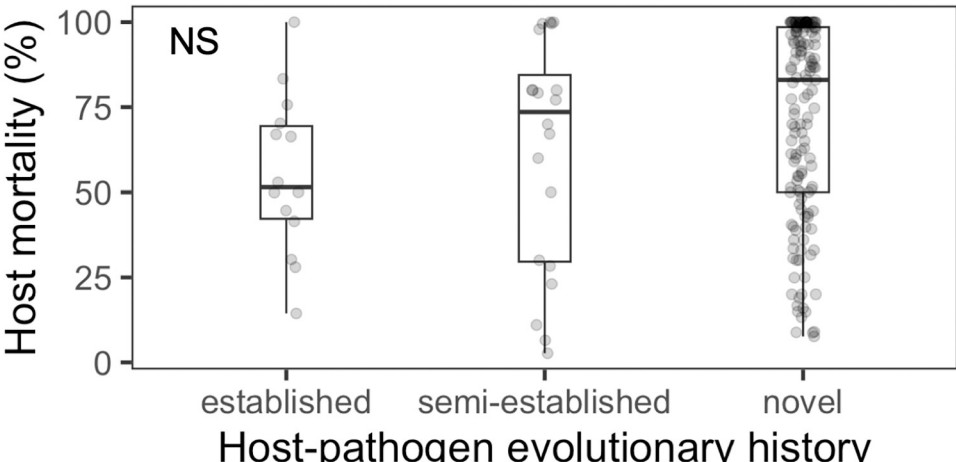

**Fig 2.** Host killing at (A) low/baseline temperatures and (B) high temperatures, for host–pathogen systems with different evolutionary histories. Under low/baseline temperatures (A), significantly higher host killing is observed in "novel" compared to "established" systems. The data and code needed to generate this figure can be found in https://doi.org/10.6084/m9.figshare.22060646.v7.

change (S5A Fig, novel:temperature_span es = 0.30, 95% CI [0.07, 0.52], $p$ = 0.009; semi-established:temperature_span es = 0.36, 95% CI [0.11, 0.60], $p$ = 0.004). This finding was further supported by a re-analysis of a single pathogen species—*Beauveria bassiana*, comprising 25 effect sizes (all from novel/semi-established systems), which demonstrated that higher level of host mortality during infection by this fungal pathogen was associated with greater temperature changes (S5B Fig, $p$ = 0.0011).

Experimental temperature increases included in our analysis ranged from 3˚C to 25˚C. This range is relevant to the upper boundary of the predicted temperature rise on Earth: 4˚C increase in annual mean surface temperature by 2100 and a 14.1˚C increase by 2300 [52]. If no actions are taken to alleviate climate change, our findings suggest that the outcomes of emerging or novel infectious disease could worsen to a greater extent in regions of intense average warming and more frequent extreme thermal events [53]. Future research should explore how host responses, such as evolutionary adaptation in shorter-lived species, acclimatization in

longer-lived species, and migration strategies under warming, could mitigate these adverse effects [54].

We found little evidence that the magnitude of the temperature-host death association was influenced by host life-stage or pathogen inoculation method (S6 Fig and S4 Table, $p > 0.05$ for all individual and interactive effects). We did not find a consistent effect of exposure time on warming-induced changes in host mortality, across host–pathogen systems (S4 Table, hypothesis 2). Other moderators, such as host immune complexity and host source, were excluded from the model selection process due to their low importance and wide confidence intervals that spanned zero (S4 Fig and S4 Table). However, a significant interaction was observed between pathogen dosage and host–pathogen evolutionary history (S7 Fig, dosage:semi-established es $> 0$, $p = 0.007$), suggesting that higher pathogen dosage was associated with increased host mortality in semi-established systems under warming, compared to established systems.

## Direction of temperature change is vital for fungal infection outcomes

We hypothesized that warming-induced increase in host mortality observed in certain subgroups, such as bacterial infections, may be driven by faster pathogen growth or higher within-host loads. To test this hypothesis, we estimated pathogen thermal optima ($T_{opt}$) for 39 well-studied species, based on their in vitro growth rate data (S3 Table). However, our analysis showed no significant effect of temperature change directions on effect sizes for bacterial infections (S4 Table, $p = 0.4626$). Expanding the analysis to other pathogen types, we found that the direction of the temperature shift relative to $T_{opt}$ was vital for outcomes of fungal infections (Fig 3, Towards $T_{opt}$ es $= 0.97$, $p = 0.037$). Unlike bacterial pathogens, fungi generally did not cause higher host mortality under warmer temperatures (Fig 1B). This outcome supported the observation that fungi are more heat-sensitive than bacteria [55]. However, host death caused by fungi increased when the temperature was shifted towards fungal $T_{opt}$ (Fig 3). Notably, as 92% of fungal effect sizes were calculated from novel host–fungi systems, our results might help predict patterns of emerging fungal diseases. Fungal species often have limited growth at elevated temperatures which constrains their ability to infect mammals [56]. However, our finding suggests that if temperatures shift towards pathogen $T_{opt}$, fungi could become more virulent. These results are consistent with observations that *Batrachochytrium dendrobatidis* (Bd)—a deadly fungal pathogen driving the extinction of amphibian species—is more prevalent at highland localities where temperatures approach the pathogen's $T_{opt}$ [57]. At higher latitudes, persistently warmer temperatures caused by climate change are contributing to the ongoing expansion of fungal pathogens such as *Coccidioides*, *Blastomyces*, *Histoplasma*, and *Sporothrix* [58,59]. With continued warming in these regions, emerging fungal infections could spread faster and be more severe, posing a greater threat to endangered ectotherms including amphibians and reptiles [60].

Host thermal tolerance might also play an important role in shaping infection outcomes under warming conditions [31]. Unfortunately, we were unable to obtain data on host thermal optima primarily due to lack of experimental evidence. Host immune response is another critical factor. For example, host immunopathology induced by infection can be heightened under elevated temperatures, causing host cell and tissue damage and worsening infection outcomes [61,62]. It can be difficult to distinguish the adverse effects of host immunopathology from direct effects of pathogens under warming [63]. Given these complexities, further experiments on the impact of temperature on immune activation and immunopathology are needed for disentangling the host and pathogen-imposed adverse effects. Tackling the impact of global climate change on wildlife health requires an understanding of the mechanisms that might increase the likelihood of host mortality when periods of extreme warming occur.

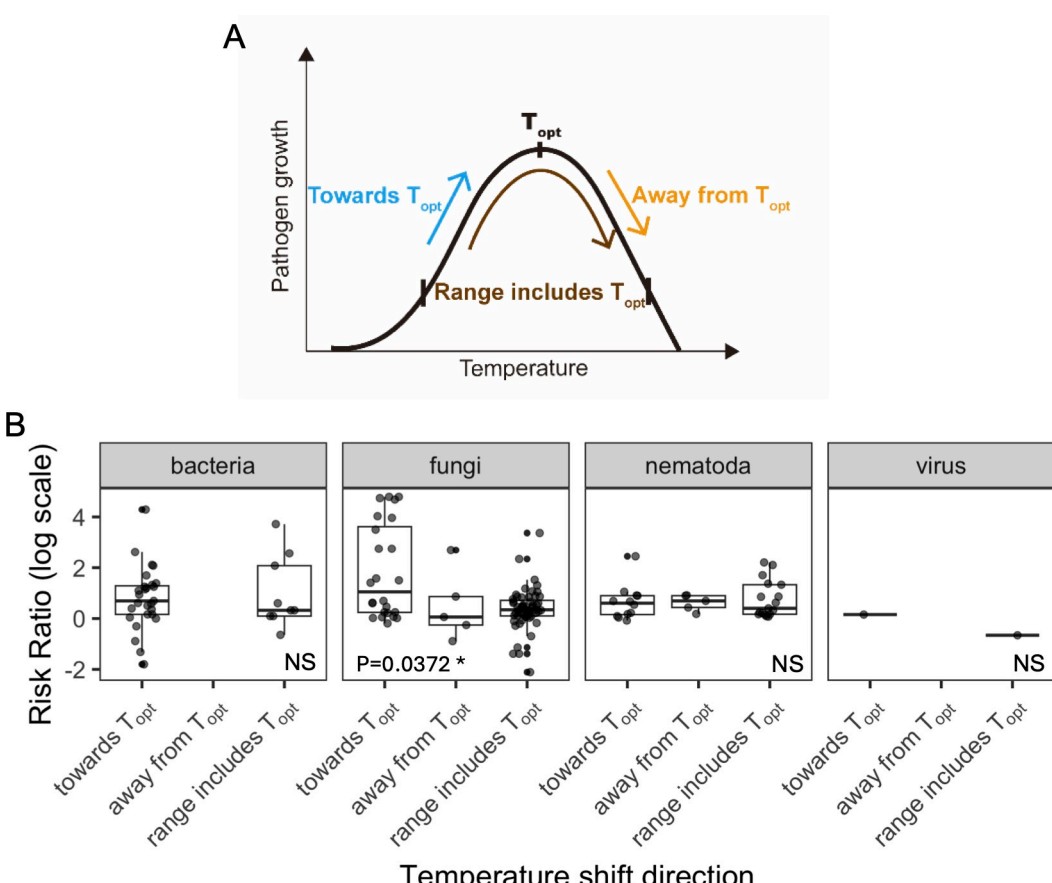

**Fig 3. RR varied with direction of temperature change.** (A) Schematic illustration of the classification for directions of temperature shift related to pathogen thermal optima ($T_{opt}$). The curve represents a generic thermal performance curve with $T_{opt}$ as the optimal growth temperature. The direction of a temperature shift is represented by arrows. "Towards $T_{opt}$": the temperature range in the study was below $T_{opt}$. "Away from $T_{opt}$": the temperature range was above $T_{opt}$. "Range includes $T_{opt}$": the low and high temperatures in the study straddled $T_{opt}$. (B) Comparisons between effect sizes across temperature shift directions, within each pathogen type. Individual effect sizes are displayed as jittered points. Effect sizes for fungal pathogens have significant larger increase from "away from $T_{opt}$" to "towards $T_{opt}$" ($p = 0.0372$). The data and code needed to generate this figure can be found in https://doi.org/10.6084/m9.figshare.22060646.v7.

## Conclusions and future perspective

Our study underscores the need for a broader range of host–pathogen systems to be included in future experimental tests of the temperature–virulence relationship. The current research landscape is skewed, while invertebrate hosts included in our analysis ranged from terrestrial (e.g., insects, mite, etc.; 78.1% of the effect sizes) to aquatic (e.g., corals, oysters, etc.; 12% of the effect sizes), there were fewer experimental studies involving aquatic vertebrate hosts (fishes, amphibians, 9.9% of the effect sizes) and none with terrestrial vertebrates. This bias is likely caused by a high prevalence of studies related to agriculture (e.g., used Arthropoda hosts) or aquaculture (e.g., used fish hosts). Our study focused on ectotherms in which both the host and pathogens are directly exposed to warming. Whether endothermic hosts—with more complex thermal regulation mechanisms—have similar responses requires further study. Indeed, most emerging human pathogens originate in other vertebrates, especially mammals [64–66]. Further research on the impact of higher temperatures in vertebrate–pathogen systems is essential for robust predictions on the outcomes of infections with zoonotic potential under climate change.

Both ecological and evolutionary perspectives are essential for understanding the impacts of rising temperatures on ecosystems. Our study revealed the effect of temperature on infection outcomes within a host's lifetime. Over longer evolutionary timescales, the difference between host and pathogen ability to adapt with thermal changes may be increasingly important to infection outcomes. Shorter generation times might help pathogens adapt faster to temperature shifts than their hosts [67], leading to increased pathogen growth within hosts. To test whether the pattern observed on ecological timescales will remain the same across generations, longitudinal studies (e.g., experimental evolution) could be used to track the potential for host and pathogen adaptation under temperatures relevant to climate change projections. Slow temperature increases may accommodate host acclimation (in ecological time) or adaptation (in evolutionary time), potentially mitigating the negative effect of warming on infection outcomes [68].

Parasites and pathogens are ubiquitous. It is thus critical to understand how warming might impact virulence to better understand the consequences for animal species during global climate change [69–73]. Our meta-analysis provides evidence that warming, and the degree and direction of temperature increase, can impact mortality in infected animal populations. We took a conservative approach by only focusing on host mortality. However, if morbidity were also considered, there might be more widespread deleterious effects of warming on infected animals. Our findings highlight distinct consequences of warming on bacterial and fungal infections, as well as for hosts in different habitats. If concomitant with bacterial pathogen epidemics, warming could exacerbate aquatic animal population declines. While if temperatures create more optimal conditions for pathogen growth, emerging fungal diseases in terrestrial animals are more of a concern. There are important consequences in this temperature–virulence relationship for species at risk and ecosystems sensitive to biodiversity loss. Human activities are warming the Earth at an unprecedented rate [74] and leading to an increased frequency and intensity of temperature extremes [52]. The current global effort to restrict warming to 1.5˚C above pre-industrial levels by 2050 [52] may also have the added benefit of limiting the lethal impacts of infectious diseases.

## Supporting information

**S1 Text. (Word document).** List of studies included PRISMA-EcoEvo Checklist.
(DOCX)

**S1 Fig. PRISMA-style flow diagram of screening and inclusion of journal articles.** The 1,649 records from a Web of Science search were screened based on our criteria, resulting in the 60 total articles included in the meta-analysis; *n* represents the number of articles.
(PDF)

**S2 Fig. The evolutionary history of the 101 host–pathogen combinations included in the meta-analysis.** Each is colored by the evolutionary history **of** the host–pathogen system (established, semi-established, and novel as defined in the main text). The data and code needed to generate this figure can be found in https://doi.org/10.6084/m9.figshare.22060646.v7.
(PDF)

**S3 Fig. Geographical locations of pathogens with known collection sites.** Some pathogens were collected years ago and may have adapted to lab environment. Different colors represent pathogen taxa (shown on genus-level when available). Collection sites data were variable in their geographic specificity, ranging from precise coordinates to broader regions such as cities and countries, depending on details reported in the original literature. The proportions in each pie chart represents proportion of pathogen taxa from each collection site. Map lines

delineate study areas and do not necessarily depict accepted national boundaries. The data and code needed to generate this figure can be found in https://doi.org/10.6084/m9.figshare.22060646.v7. Base layer of the map is from the Natural Earth 1:50 m map (https://www.naturalearthdata.com/), under a non-exclusive license.
(PDF)

**S4 Fig. Model-averaged importance of moderators.** Moderators including "host immune complexity" and "host collected field or lab" with low importance values (model-averaged importance <0.25 and 95% CI included zero) were considered to have little support for inclusion in the model. The data and code needed to generate this figure can be found in https://doi.org/10.6084/m9.figshare.22060646.v7.
(PDF)

**S5 Fig. Association between the span of temperature change and effect sizes.** (A) Temperature change is interacting with host–pathogen evolutionary history to influence effect sizes. (B) Increases in effect sizes are associated with larger temperature changes across studies of the pathogen Beauveria bassiana (k: 25 effect sizes across multiple hosts). The data and code needed to generate this figure can be found in https://doi.org/10.6084/m9.figshare.22060646.v7.
(PDF)

**S6 Fig.** Effect sizes were not impacted by (A) host life-stage, or (B) pathogen was inoculated by injection or other means. Individual effect sizes are displayed as jittered points. The data and code needed to generate this figure can be found in https://doi.org/10.6084/m9.figshare.22060646.v7.
(PDF)

**S7 Fig. The impacts of pathogen dosage on effect sizes.** Positive interaction effect between pathogen dosage and semi-established system is observed. The data and code needed to generate this figure can be found in https://doi.org/10.6084/m9.figshare.22060646.v7.
(PDF)

**S1 Table. Summary of included studies and collected moderator variables.**
(XLSX)

**S2 Table. Effect sizes calculated for meta-analysis.**
(XLSX)

**S3 Table. Pathogen in vitro $T_{opt}$ data collected from published literatures.**
(XLSX)

**S4 Table. Results from modeling and statistical analyses.**
(XLSX)

## Author Contributions

**Conceptualization:** Jingdi Li, Georgia C. Drew, Justyna Wolinska, Kayla C. King.

**Data curation:** Jingdi Li, Nele Guttmann, Georgia C. Drew.

**Formal analysis:** Jingdi Li.

**Funding acquisition:** Justyna Wolinska, Kayla C. King.

**Investigation:** Nele Guttmann.

**Methodology:** Jingdi Li.

**Supervision:** Justyna Wolinska, Kayla C. King.

**Validation:** Jingdi Li, Georgia C. Drew, Tobias E. Hector, Justyna Wolinska, Kayla C. King.

**Visualization:** Jingdi Li.

**Writing – original draft:** Jingdi Li, Kayla C. King.

**Writing – review & editing:** Jingdi Li, Nele Guttmann, Georgia C. Drew, Tobias E. Hector, Justyna Wolinska, Kayla C. King.

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
