## [Editor Report · Decision Letter 0]

14 May 2024

Dear Dr Li, 

Thank you for submitting your manuscript entitled "Warming-induced excess deaths of infected ectotherms depend on pathogen kingdom and evolutionary history" for consideration as a Meta-Research Article by PLOS Biology.

Your manuscript has now been evaluated by the PLOS Biology editorial staff, as well as by an academic editor with relevant expertise, and I'm writing to let you know that we would like to send your submission out for external peer review.

IMPORTANT: We think that your study would be better considered as a Short Report. No re-formatting is required, but please change the article type to "Short Reports" when you upload your additional metadata (see next paragraph).

Once your full submission is complete, your paper will undergo a series of checks in preparation for peer review. After your manuscript has passed the checks it will be sent out for review. To provide the metadata for your submission, please Login to Editorial Manager (https://www.editorialmanager.com/pbiology) within two working days, i.e. by May 16 2024 11:59PM.

Kind regards,

Roli Roberts

Roland Roberts, PhD

Senior Editor

PLOS Biology

rroberts@plos.org

---

## [Decision Letter · Decision Letter 1]

25 Jul 2024

Dear Dr Li,

Thank you for your patience while your manuscript "Warming-induced excess deaths of infected ectotherms depend on pathogen kingdom and evolutionary history" was peer-reviewed at PLOS Biology. It has now been evaluated by the PLOS Biology editors, an Academic Editor with relevant expertise, and by two independent reviewers. 

You'll see that reviewer #1 is very positive about the study, and has largely textual requests. Reviewer #2 is also broadly positive, but wants more detail and significant improvements in your analysis; specifically, s/he wants you to adjust for host and pathogen phylogeny, fit a single joint model (rather than multiple independent ones), and move away from p-values and significance testing. s/he also wants you to discuss sub-lethal effects of temperature on hosts, and has a series of other textual points. During cross-commenting, reviewer #1 concurred with reviewer #2's points.

After discussing the reviews with the Academic Editor, we would like to invite you to revise the work to thoroughly address the reviewers' reports.

Given the extent of revision needed, we cannot make a decision about publication until we have seen the revised manuscript and your response to the reviewers' comments. Your revised manuscript is likely to be sent for further evaluation by all or a subset of the reviewers.

**IMPORTANT - SUBMITTING YOUR REVISION**

*Re-submission Checklist*

*Published Peer Review*

*PLOS Data Policy*

*Blot and Gel Data Policy*

Sincerely,

Roli Roberts

Roland Roberts, PhD

Senior Editor

PLOS Biology

rroberts@plos.org

REVIEWERS' COMMENTS:

Reviewer #1:

This meta-analysis tackles an overlooked potential complication of climate change: worsening disease severity in animals. It is based on a data set across ectotherms. It nicely takes a very conservative approach, focusing only on host death. This means that the effect of raising temperature is likely even greater if measures of morbidity had been included. The introduction frames the broader question and provides a quick review of possible mechanisms for why increased temperature can worsen disease. 

The search terms and inclusion criteria were thoughtfully chosen and defended. I especially liked that they excluded unnatural temperature extremes that would likely bias the outcomes. The handling of death in controls and choice of data collection point and the inclusion of an evolutionary history designation were both great. Moderator variables were also thoughtful. 

Minor corrections/suggestions

Line 86. Novel transmission opportunities? Or do you mean host jumping?

Line 88. Alter or reduce? You mean reduce, right if the second half of the sentence is true. Then be specific. 

Differentiate the following two statements better. Confusing as written also as some of this is counter to expectation. 

Line 287 We found that the effects of excess host death in established, but not novel systems were the most pronounced in host-bacterial pathogen systems 

And

293 We also found that host mortality in novel systems was already higher than that in established associations, at low/baseline temperature 

In the discussion you should highlight that if morbidity rather than just mortality was included, effects will be even more widespread. 

Reviewer #2:

Overall thoughts

This is a very interesting an timely meta-analysis of the role of temperature change on infection-induced mortality. I found the introduction and the manuscript in general to be well-written and a pleasure to read. While the results are interesting, I found the statistical methods in particular to be lacking in detail, and overly reliant on p-values and significance testing. It is difficult to discern the exact models fit and examined from the Main MS, and the Supplementary tables do not clearly outline the different models. Further, the authors do not appear to have adequately adjusted for host or pathogen taxonomy, and as I can see they do not fit any hierarchical effects, which would be appropriate given the multiple levels of replication in their data. 

Major points

- Line 221-224: It appears that you have not adjusted for host taxonomy or phylogenetic relationships among taxa other than gross pathogen type, and host Phylum as binary (C

hordata or not). Phylogenetic or taxonomic non-independence is an important aspect to consider in your models. If phylogenetic relationships among your taxa are unavailable, you may include taxonomic non-independence in these hierarchical effects. For example, a taxonomic tree could be easily obtained and converted to a tree with 1 unit distance associated to each taxonomic level (e.g. species to genus, genus to family, family to order, etc...). I strongly suggest you include this as a structuring factor for your species-level hierarchical effects. Further, you may follow the appraoch first introduced by Schmidt et al (2021) (https://onlinelibrary.wiley.com/doi/abs/10.1111/ele.13740), and later shown to be best practice recommended by Cinar et al (2022) (https://besjournals.onlinelibrary.wiley.com/doi/full/10.1111/2041-210X.13760) of including both taxonomically structured and non-taxonomic structured hierarchical effects for both host and pathogen, as using only the Any attempts to simplify this model, such as using only one of these components may lead to erroneous inferences from the data.

- Line 225-231: It is unclear why you need to do this sub-group analyses rather than fitting a single joint model. For these types of hierarchical models, it is generally considered best practice to fully and jointly model all levels of the data simultaneously. Please provide additional detail on why this was necessary and what was learned compared to fitting a fully hierarchical model. Currently it is unclear from the MS alone what your main model was. Instead, it appears that you conducted multiple independent models.

- While I do not support a p-value / "significance" based paradigm of assessing predictors, you do not appear to correct or discuss the issue of multiple model fitting and p-value correction. I suggest you re-do analyses to fit a main joint model, with subsequent sensitivity models, and discuss relative effect sizes and credible / confidence intervals rather than assessing importance based on p-values alone.

- While you frame temperature changes with respect to pathogen thermal optima, you do not discuss host thermal optima in this context. While you adjust for host mortality due to warming, there is little to no consideration of sub-lethal changes in host responses to shifting temperatures. Further, I cannot find mention of the extent to which hosts or pathogens are pre-acclimated at their Topt or to another condition before the experiments took place. Considering this is an important part of the framing of your discussion, it would be good to clarify this in the methods. 

Minor points

- Lines 134-137: For reproducibility and broader context it would be good to state the institution from which the search was performed as this impacts the results of WOS searches (see https://gehmana.weebly.com/uploads/3/1/9/8/31987469/dallas_gehman_farrell_2018.pdf). Also, I see that the numbers of articles returned by your search, and retained in the meta-analysis are in the SI, but for clarity please also include these numbers in the main manuscript. 

- Lines 148-158: Unclear why all experimental conditions were not included in analyses. The idea that omitting potential relevant data to "maximise the ecological relevant" is not well explained. You could look at the relative difference in temperature between all time points in a study, which would give you some larger statistical power to describe any effects. Any removal of data should be clearly explained and justified, or else this feels like an arbitrary choice.

- Line 165-166: Please define what "large host mortality" is. Is this above 10%, or some other proportion which was too large to correct for with Abbot's formula?

- Line 186-187: Were there any cases in which the the study temperature was significantly reduced from T opt? In this case, wouldn't "Away from T opt" not always include temperatures above the T opt? I assume you are always ordering the experimental conditions to reflect a warming of temperature, but in some cases, I expect the experimental conditions may be structured to reflect a progressive cooling of temperature. In either case, some clarification of this would be nice.

- Line 212-214: Why was a p>0.07 threshold used here?

- Line 249-250: Why do you think your results support a diminishing of host tolerance with increased temperature, rather than the other competing hypotheses you outlined in the introduction?

- Line 286-288: Since this interaction was found to be significant, it would be best to plot this result in the main MS rather than the effects of individual predictors ignoring this effect.

- Line 293-299: The finding that higher baseline mortality was found in novel systems is a very interesting (and core) finding! Considering this, I highly suggest you generate a plot showing this difference in baseline mortality for the main MS. Also, some numeric quantification of the difference in baseline rates would be nice to see. 

- Line 353-356: For fungi, the vast majority of experiments were with "novel" host-pathogen combinations, which you state previously as having larger baseline mortality rates. Could this be causing your lack of effect? In this case, are you justified in assuming that fungi do not result in higher host death under warming conditions?

---

## [Decision Letter · Decision Letter 2]

4 Oct 2024

Dear Dr Li,

Thank you for your patience while we considered your revised manuscript "Warming-induced excess mortality of infected ectotherms depend on pathogen kingdom and evolutionary history" for consideration as a Short Reports at PLOS Biology. Your revised study has now been evaluated by the PLOS Biology editors, the Academic Editor and the original reviewers.

In light of the reviews, which you will find at the end of this email, we are pleased to offer you the opportunity to address the remaining points from the reviewers in a revision that we anticipate should not take you very long. We will then assess your revised manuscript and your response to the reviewers' comments with our Academic Editor aiming to avoid further rounds of peer-review, although might need to consult with the reviewers, depending on the nature of the revisions.

IMPORTANT - Please attend to the following:

a) Please change your Title slightly to make it easier to parse: "Excess mortality of infected ectotherms induced by climate change depends on pathogen kingdom and evolutionary history"

b) We find the following passage in the Abstract a little hard to follow: "we demonstrate that warming significantly increased the mortality of hosts infected by bacterial pathogens, as well as pathogens established within the host species." Maybe it would be clearer to separate out these two findings?

c) Please address the remaining concerns raised by reviewer #2.

d) Please ensure that you comply with our Data Policy; specifically, we need you to supply the numerical values underlying Figs 1ABC, 2AB, 3B, S2, S3, S4, S5AB, S6AB, S7, either as a supplementary data file or as a permanent DOI’d deposition. I note that you already have an associated Figshare deposition. Please could you clarify whether this contains the data and code needed to recreate the Figures?

e) Please cite the location of the data clearly in all relevant main and supplementary Figure legends, e.g. “The data underlying this Figure can be found in S1 Data” or “The data and code needed to generate this Figure can be found in https://doi.org/10.6084/m9.figshare.22060646.v6"

f) Please make any custom code available, either as a supplementary file or as part of your data deposition.

**IMPORTANT - SUBMITTING YOUR REVISION**

*Resubmission Checklist*

*Published Peer Review*

*PLOS Data Policy*

*Blot and Gel Data Policy*

Sincerely,

Roli Roberts

Roland Roberts, PhD

Senior Editor

PLOS Biology

rroberts@plos.org

REVIEWERS' COMMENTS:

Reviewer #1:

[checked "Accept" with no further comments]

Reviewer #2:

Major comments / concerns

I reviewed a previous submission of this manuscript, and am very happy to see that the authors have address the majority of my methodological concerns, and incorporated my modelling suggestions into the manuscript. However, it is not recommended to exclude hierarchical effects, even if they appear to explain little variation in the model (lines 2412-245). As per Cinar (2022), it is vital to include both taxonomic and non-taxonomic effects in these models to account for non-independence. If they turn out to explain very little variation, then this is fine , but they should still be included in any models. 

While I personally think is not justified to do model selection or model averaging when you have clear hypotheses (which you do!), if you choose to go forward with your model selection approach (which I also highly recommend against doing), it would be considered statistical best practice to force these four taxa-level hierarchical effects and the study-level hierarchical effect in your "final" model(s). 

After addressing this, one major issue methodological remains - the authors have not adequately justified their choice to "maximise the ecological relevance" of the study by excluding low and high temperatures and pathogen inoculation doses per study (Lines 145-161). This was a concern in my previous review, and the authors responded by stating the aim of the study was to "capture the most substantial changes in host fitness by selecting conditions with the largest temperature changes, while avoiding confounding effects of extreme temperatures and dosages". 

The authors' justification for excluding most substantial changes makes sense, but is not justified through any analysis or presentation of data. This justification assumes that extreme values are those not experienced by hosts or pathogens in natural settings, but the authors say in their response letter that do not know know the thermal optima for hosts, meaning that you do not know the range of temperatures that would be considered "extreme". In this case I feel they need to show that these extreme temperatures are associated with ill effects on hosts, and this if course would be host specific.

Without data to support it, the authors state in both the review response and the manuscript that in experimental studies the lowest and highest temperatures often represent conditions that are rarely encountered by hosts and pathogens in natural settings, and that these typically result in host death in the absence of pathogen exposure. However, because the authors have stated that they "ensured that all host mortality in the pathogen treatment groups was due to infection rather than other factors", they should be able to include some of the more extreme temperature effects if no host mortality was observed. 

This may seem minor, but is concerning to me that readers may assume the authors have made some arbitrary choices to filter their data to identify a clear effect. The goal of the study is to look at extreme impacts of temperature and pathogen infection, but the authors exclude many of these "extreme" treatments without explicitly defining how extreme these are (e.g. they remove high and low treatments per study without quantifying how extreme these are, or how extreme they are with respect to "natural" conditions"). If this is not possible, a "sensitivity analysis" including all treatments could be included in the supplement as a comparison.

Minor comments / concerns

- Line 89: I'm not sure what "Higher virulent temperatures" means - do you simply mean "higher temperatures"?

- Line 99: Grammatical error: "changes" should be "change"

- Lines 131-133: You describe the search terms as "keywords", which is confusing. It is unclear which categories of the WOS Core Collection you are searching with these terms. Did you restrict to searching paper keywords with your search terms, or did you use abstracts, or titles, or the proprietary "Topic" field in WOS? This is important for reproducibility, and clarity of your methods. Further, you say you uses two search strings, which were "paired". By "paired", I assume you are linking these terms via an AND statement (e.g. "'parasit*' OR 'pathogen*' AND 'temperature' AND 'virulen*'). Please clarify the exact search string you used.

- Line 237-238: Thank you for incorporating both taxonomic and non-taxonomic effects. As this is a somewhat new best practice for comparative models, please cite both Schmidt et al (2021) (https://onlinelibrary.wiley.com/doi/abs/10.1111/ele.13740), and Cinar et al (2022) (https://besjournals.onlinelibrary.wiley.com/doi/full/10.1111/2041-210X.13760) to give proper credit.:

Something like "as first proposed in Schmidt et al. (2018) and later shown to be best practice by Cinar et al. (2022)", would be fantastic - thank you.

---

## [Editor Report · Decision Letter 3]

14 Oct 2024

Dear Dr Li,

Thank you for the submission of your revised Short Reports "Excess mortality of infected ectotherms induced by warming depends on pathogen kingdom and evolutionary history" for publication in PLOS Biology. On behalf of my colleagues and the Academic Editor, Lauren Buckley, I'm pleased to say that we can in principle accept your manuscript for publication, provided you address any remaining formatting and reporting issues. These will be detailed in an email you should receive within 2-3 business days from our colleagues in the journal operations team; no action is required from you until then. Please note that we will not be able to formally accept your manuscript and schedule it for publication until you have completed any requested changes.

Sincerely, 

Roli Roberts

Senior Editor

PLOS Biology

rroberts@plos.org